# Breastfeeding Practice and Association between Characteristics and Experiences of Mothers Living in Bangkok

**DOI:** 10.3390/ijerph18157889

**Published:** 2021-07-26

**Authors:** Chompoonut Topothai, Thitikorn Topothai, Rapeepong Suphanchaimat, Walaiporn Patcharanarumol, Weerasak Putthasri, Yupayong Hangchaowanich, Viroj Tangcharoensathien

**Affiliations:** 1International Health Policy Program, Ministry of Public Health, Nonthaburi 11000, Thailand; thitikorn@ihpp.thaigov.net (T.T.); rapeepong@ihpp.thaigov.net (R.S.); walaiporn@ihpp.thaigov.net (W.P.); viroj@ihpp.thaigov.net (V.T.); 2Bureau of Health Promotion, Department of Health, Ministry of Public Health, Nonthaburi 11000, Thailand; 3Division of Physical Activity and Health, Department of Health, Ministry of Public Health, Nonthaburi 11000, Thailand; 4Division of Epidemiology, Department of Disease Control, Ministry of Public Health, Nonthaburi 11000, Thailand; 5National Health Commission Office, Nonthaburi 11000, Thailand; weerasak@nationalhealth.or.th; 6Charoenkrung Pracharak Hospital, Bangkok 10120, Thailand; dryupayong@gmail.com

**Keywords:** breastfeeding, determinant, Bangkok, maternal characteristic, health services

## Abstract

Although the benefits of breastfeeding are widely recognized, only 14% of mothers in Thailand exclusively breastfed their children during the first six months of their lives in 2019, which dropped from 23% in 2016. This study aimed to assess the prevalence of exclusive breastfeeding (EBF) up to six months, current breastfeeding patterns, and key determinants that influence six-month EBF among mothers residing in Bangkok, Thailand. A cross-sectional study was conducted using a self-administered questionnaire survey. In total, 676 healthy mothers living in Bangkok, whose most recent child was between 6 and 18 months old, were recruited. Descriptive statistics, univariable analysis by Chi-square test, and multivariable logistic regression were performed to assess the association between six-month EBF and maternal characteristics and experiences of using maternal health services. The prevalence of six-month EBF of infants in Bangkok was 41%. The key determinants that influenced six-month EBF included: maternal age of more than 30 years; higher education level; higher maternal income; multi-parity; exposure to breastfeeding advice during pregnancy; intention to breastfeed for a long duration (≥6 months) during pregnancy; experience of six-month EBF in the previous child. This study draws health professionals’ and policy makers’ attention to further promote breastfeeding in particular types of mothers.

## 1. Introduction

The benefits of breastfeeding for both children and mothers are well-known [1]. Breastmilk not only protects infants from infectious diseases, especially diarrhoea and pneumonia but also provides supreme nutrients for infants’ growth and development. Breastfed infants face lower risk of type 2 diabetes and obesity, and are likely to enjoy better promotion of cognitive development in the long-term [1]. Lactating mothers who continue breastfeeding over a longer duration (more than 12 months) are less likely to develop breast and ovarian cancers [2]. Furthermore, the outstanding advantages of breastfeeding are evident by the economic benefits to the mother, and positive environmental impacts [3].

The World Health Organization (WHO) and the United Nation Children’s Fund (UNICEF) suggest that all infants should be initially breastfed within one hour after delivery, exclusively breastfed to six months, and breastfed for two years or beyond with appropriate complementary feeding [4]. This recommendation is linked to the global target to promote exclusive breastfeeding (EBF) rate in the first six months to achieve a minimum rate of 50% by 2025 and 70% by 2030 [5].

Global evidence suggests that breastfeeding is a multifactorial practice which is influenced by maternal profiles, country’s health systems and socio-economic context, and market promotion activities by breast milk substitute industries [3]. Maternal characteristics, such as age, education level, parity, employment, and place of residence and living standards, are identified as key predictors for breastfeeding initiation and continuation [6,7,8,9]. Breastfeeding outcomes also depend on support from within the health system, especially the support for early initiation of breastfeeding, uninterrupted skin-to-skin contact, and rooming-in through the implementation of the Baby-Friendly Hospital Initiative (BFHI) as recommended by WHO [10]. Health professionals are recognized as key people who can influence optimal breastfeeding through counselling and nursing care [8,11,12]. Other determinants associated with breastfeeding include husband and family support, the workplace environment, socio-cultural norms, and the influence of infant formula marketing [3,13].

The Royal Thai Government has announced a national breastfeeding recommendation and target in line with the global community goals [14,15], yet, the outcome is still disappointing. The EBF rate of infants under 6 months, using the 24-h recall period, was only 14% in the latest Multiple-Indicator Cluster Survey (MICS) conducted in 2019 [16]. This dropped sharply from 23% in 2016 [17], despite its increasing trend since 2006 [18,19]. Inadequate breastfeeding practice in Thailand resulted in significant losses for the nation since a study in 2015 estimated that the cost of inadequate breastfeeding may exceed 0.5% of Gross National Income in Thailand [20].

To further develop effective interventions for promoting optimal breastfeeding in Thailand, it is important to identify the current breastfeeding pattern of mothers and the factors that affect their feeding choices. This study, therefore, aims to assess the prevalence of 6-month EBF practice, current breastfeeding patterns, and the association with maternal characteristics and experiences of using health services among Thai mothers living in Bangkok. Bangkok is chosen as a study site as it is a metropolitan area where mothers are exposed to highly urbanized lifestyles and high female labor participation rate. Labor force participation was 33% among female 15–24 years old in 2020 [21]. In addition, the rate of EBF of infants aged 0–5 months living in Bangkok was extremely low and below the national average for the past ten years, while the number of child births per year was the highest [17,18,19]. Additionally, prior studies have not yet explored the EBF patterns of mothers in Bangkok and factors associated with EBF outcomes [22,23,24,25,26,27]. This study can fill the knowledge gaps regarding EBF patterns in a metropolitan area. Lessons are useful for other low-and middle-income (LMIC) metropolitan settings where mothers face similar lifestyles.

## 2. Materials and Methods

### 2.1. Study Design, Population, and Sample Size

We conducted a study in Bangkok, Thailand. A cross-sectional survey was performed. Participating mothers needed to follow the following inclusion criteria: they were healthy mothers living in Bangkok and their most recent child was aged 6–18 months who had a birth weight of more than 2500 g. The sample size calculation was based on the formula *n* = Z^2^_α/2_ PQ/d^2^ where the prevalence of EBF (P) for infants under 6 months equated to 23% (according to the latest national survey at the time of project development). The level of significance of 5%, design effect of 2, and estimated 20% non-response rate were taken into account. In total, the estimated sample size was 660.

A two-stage sampling technique was applied for participant selection. The first stage was to select health facilities in Bangkok that provided routine health check-ups and vaccinations for children aged 0–6 years old (well-child clinics). There were 105 health facilities in Bangkok that met the criteria; 69 (66%) primary health centers and 36 (34%) hospitals. In total, 22 out of 105 health facilities were proportionately selected with 14 (64%) primary health centers and 8 (36%) hospitals. The maternity services provided to mothers and children by hospitals and primary health centers may be different in terms of delivery services and the design of parental class and counselling programs. All primary health centers have no labor room and cannot provide delivery services. They also have no separate rooms for parental counselling, thus, health professionals usually give advice to each individual family. A majority of hospitals arrange parental classes for mothers as a separate activity that can be held either for an individual or a group. The second stage was the selection of individual mothers who brought their children to the well-child clinic in the selected health facilities. At the beginning, the research team expected to recruit mothers in a facility based on a quota sampling: 10–20 respondents per each primary health center and 40–50 respondents per each hospital. However, in practice, the number of mothers obtaining services was very different in each site, even among the same types of health facilities. The full list of hospitals and health centers with a corresponding number of samples acquired is presented in Appendix A. The participants were approached at the registration desk of the well-child clinic and invited to participate in the study. If they were interested in the project and gave written consent, they proceeded to the self-administered questionnaire.

### 2.2. Questionnaire Development

The questionnaire was developed by adapting a framework and existing knowledge from the prior literature [3,22]. Then, it was reviewed and revised by three breastfeeding experts involved in breastfeeding research, program development, and policy implementation. The questionnaire was then piloted with 30 mothers attending the well-child clinic in the Metropolitan Health and Wellness Institution located in Bangkok, in order to validate the questionnaire’s clarity. Finally, the questionnaire composed of four main parts: (i) demographic data of mothers and children; (ii) experiences in health services and support related to breastfeeding during prenatal and postnatal periods; (iii) breastfeeding experiences; (iv) maternal employment and workplace environment. The key variables included in the questionnaire were: age (years), education level, maternal monthly income (Baht), experience of EBF for the previous child (yes vs no), exposure to breastfeeding advice during pregnancy period (yes vs no), intention to breastfeed during pregnancy period (months), mode of infant delivery (normal labor vs Caesarean section vs labor assisted), exposure to breastfeeding problems after delivery (yes vs no), and working status (jobless vs currently working).

### 2.3. Data Collection

The data were collected between February and November 2020. There were two teams of data collectors. The first team comprised 2 researchers and the other team comprised 7 trained data collectors. The two researchers collected data at the hospitals while the trained data collectors collected data at primary health centers. Each respondent took approximately 15 min to complete a questionnaire.

### 2.4. Variable Management

The primary outcome variable of the study was the prevalence of EBF practice during the first six months. Mothers achieving 6-month EBF were defined as EBF mothers, while those who did not were defined as non-EBF mothers. The EBF practice was analyzed based on data retrieved by following three key questions: did you ever breastfeed your child?; do you still breastfeed your child and, if not, when did you stop breastfeeding your child?; how old was your child when you introduced supplementary feeding? The third question then asked mothers to identify the first time that their children received any items of the following: water, breastmilk substitutes, complementary foods, and processed supplementary foods. By asking these questions, the EBF duration of each mother was calculated and categorized as EBF or non-EBF group.

The key independent variables were personal profiles and experiences of using health services: age group; education level; working status; maternal income range; marital status; parity; an experience of 6-month EBF in the previous child (for multiparous mothers only); an exposure to breastfeeding advice during pregnancy; an intention to breastfeed during pregnancy; mode of infant delivery; practicing of rooming-in with the infant during hospital stay; an exposure to breastfeeding problems. Chi-square was performed to assess the statistical significance in each personal attribute of EBF mothers and non-EBF mothers.

### 2.5. Data Analysis

All analyses were performed by STATA software version 14.2, StataCorp, TX, USA. (serial number 10699393). Univariable analysis by Chi-square test, was performed to assess the effect of executing EBF according to each personal attribute. We also performed multivariable analysis to assess the achievement of 6-month EBF by accounting for the influence of all covariates. We achieved this by mixed-effects multivariable logistic regression (taking the type of health facility for child vaccination at time of data collection as the higher hierarchy variable) to adjust any variations in each individual facility. The results were presented in terms of adjusted odds ratio, and 95% confidence interval (CI). The Kaplan–Meier survival analysis was applied to estimate the duration of EBF practices, and the log rank test was used to compare the median EBF duration of mothers by personal attributes.

### 2.6. Ethical Consideration

All respondents gave their informed consent before they participated in the study. At the end of the interview, each participant received a children’s book worth approximately USD 5, as a small compensation for their participation. This study received ethics approval from: (1) the Institute for Development of Human Research Protection in Thailand (IHRP letter head: 127/2563); (2) the Bangkok Metropolitan Administration Human Research Ethics Committee (Al 03.11/BMAHREC 02.1), which is a highly recognized institution that aims to protect rights, dignity, and safety and well-being of participants by conforming to the international standards, in particular the Declaration of Helsinki, Belmont Report, CIOMS Guidelines and ICH-GCP Guidelines.

## 3. Results

### 3.1. Baseline Characteristics and EBF

In total, we acquired 676 samples, slightly larger than the expected number. More than half, 59% of mothers were non-EBF, as shown in Figure 1. From the samples, 509 respondents (75.3%) were recruited from hospitals. Both primary health centers and hospitals shared the same proportion of mothers with EBF and not EBF at the ratio of approximately 2:3.

The median age was 30.0 years (p25–p75 = 24–35). Mothers with secondary education or below accounted for 59.5% of all samples. The majority of mothers (61%) were employed at the time of data collection, and the median maternal income was THB 13,000 per month (p25–p75 = THB 9000–20,000 per month). Almost all mothers (85%) lived with their husband in the same house every day. Half of the participants (51%) were first-time mothers. Among multiparous mothers, 73.2% had an experience of 6-month EBF in the previous child. During the pregnancy period, most of the mothers (82%) were exposed to breastfeeding advice, and intended to breastfeed for at least 6 months (about 78%). Almost 60% of the participants undertook normal delivery. Approximately 58% of the total participants practiced rooming-in with their babies for the whole time during the hospital stay. Approximately 57% of mothers reported that they had ever had breastfeeding problems.

Using Chi-square with *p* varied between 0.001 and 0.05, a statistical significance was found when comparing the prevalence of 6-month EBF and non-EBF mothers in the following variables: age group, education level, maternal income range, parity, an experience of 6-month EBF in the previous child, an exposure to breastfeeding advice during pregnancy, and the intention to perform breastfeeding during pregnancy. See Table 1.

Among the 541 mothers who were exposed to breastfeeding advice during pregnancy, 490 of them (90%) reported the source of advice, of which 51% were health personnel. The second and third sources of advice were relatives (26%) and friends (23%), as shown in Figure 2.

However, when experiencing breastfeeding difficulties, the majority of mothers sought support or advice from online materials (Facebook groups and pages), followed by the health service system (breastfeeding clinics and health personnel), and relatives and friends, as depicted in Figure 3.

The top three most common difficulties identified by mothers facing breastfeeding problems were insufficient milk supply, nipple wounds, and sucking problems, as demonstrated in Figure 4.

The Kaplan–Meier survival analysis was used to estimate the duration of EBF practice. The curve showed a rapid dropping of the (survival) probability of EBF at three- and six-month points of time, as depicted in Figure 5.

When comparing the duration of EBF practice by log rank test, we found that the median EBF duration in the following participant groups was significantly shorter than their counterparts; that is, at a younger age, lower education level, no experiences of achieving 6-month EBF practice in the previous child, and intention during pregnancy to breastfeed for a shorter duration (with *p* varying between 0.001 and 0.05), as shown in Table 2.

### 3.2. Participants’ Profile and EBF: Multivariable Analysis

The multivariable analysis by mixed-effects logistic regression was performed in two parts: (1) analysis on total participants and (2) analysis only on multiparous mothers.

In the first part, we initially planned to include all variables that showed statistical significance in univariable analysis in the multivariable analysis. However, the variable ‘an experience of 6-month EBF in the previous child’ was excluded as it had perfect collinearity with the variable ‘number of children’. Therefore, we selected only ‘number of children’ in the multivariable analysis. The findings revealed that mothers with intention to breastfeed for a longer duration (≥6 months) during pregnancy had greater odds of achieving 6-month EBF compared with mothers who showed shorter-duration breastfeeding intentions (OR 2.9, CI 1.7–4.9). See Table 3.

In the second part, we focused only on participants who had two or more children (N = 331). The findings are shown in Table 4. Having an experience of 6-month EBF in the previous child and showing an intention to complete a 6-month EBF during pregnancy were significantly associated with the achievement of 6-month EBF (OR 2.1, CI 1.1–4.1 and OR 3.1, CI 1.3–7.3, respectively).

## 4. Discussion

The results showed that the current EBF practice of mothers living in Bangkok, 41%, is still suboptimal. Although it is higher than the national average, this figure is still lower than 50%, as recommended by WHO. Factors associated with 6-month EBF outcomes are: maternal age, education level, income level, parity, intention to breastfeed for a longer period (≥6 months), experience of 6-month EBF in the previous child, and exposure to breastfeeding advice during pregnancy. In multivariable analysis, factors that presented with a significant association with 6-moth EBF were an experience of 6-month EBF in the previous child and an intention to complete 6-month EBF during pregnancy.

Our findings, that older mothers with a higher education and income level were more likely to successfully practice six-month EBF compared with their counterparts, are similar to prior studies in LMIC and Thailand [8,9,22,24]. Since breastfeeding demands high levels of responsibility and commitment from mothers, a younger or teenage mother normally encounters more difficulties compared with mothers of an older age [28]; younger mothers have a limited capacity to cope with unanticipated stress due to the demands and difficulties of breastfeeding [29,30]. In addition, mothers with low educational attainment may have limited understanding of, or confidence in, breastfeeding [31]. Mothers with lower-income levels may have limited access to formal and informal breastfeeding support when faced with breastfeeding difficulties [22].

The association between 6-month EBF outcome and parity is understandable as first-time mothers have limited experiences of breastfeeding and parenting, compared with more experienced mothers [32]. To facilitate successful EBF in first-time mothers, they should receive training on EBF-related knowledge and skills, and receive support to maintain their breastfeeding intentions [33,34]. The achievement of 6-month EBF practice in first time mothers is important for EBF outcomes in subsequent children [35,36].

Intention to breastfeed for a long duration during pregnancy is strongly associated with 6-month EBF in mothers with two or more children and mothers with a single child. This can be explained by the Theory of Planned Behaviour that describes that peoples’ behaviors are mainly shaped by their intention and perceived self-control [37]. Evidence also suggests that mothers who intend to breastfeed are more likely to access more sources of information and have greater knowledge about recommendations of prenatal and infant nutrition compared with mothers having little or no intention [38]. Our findings suggest that all women, once pregnant, should be empowered to develop the intention to breastfeed for at least six months. Mothers with specific characteristics, in particular, younger age, lower education and income levels, first-time mothers, and mothers with two or more children who failed to practice 6-month EBF in the previous child, are priority group for empowerment. Low-income mothers should be further supported through a specific social welfare programme, such as a child support grant, to ensure their ability to access breastfeeding support [39].

Since the exposure to health professionals’ advice on breastfeeding during pregnancy is associated with better 6-month EBF outcome, health professionals should provide adequate breastfeeding counselling and training sessions during prenatal services to pregnant women to ensure that they gain the necessary breastfeeding skills. Insufficient milk supply, nipple wounds, and poor latch-on/sucking problems are three major reasons for EBF discontinuation among Thai mothers [22,24,26]. Hence, breastfeeding counselling sessions, during both pre- and postnatal visits, should cover the techniques and skills to perform the correct latch-on position. Additionally, knowledge about breastmilk production, newborn hunger cues, and signs of sufficient breastmilk feeding should be provided to all mothers to boost their confidence in breastfeeding practice. Moreover, screening for maternal histories, attitudes, and experiences of prior breastfeeding are important for identifying and preventing the risk of early EBF cessation [40,41].

In this study, respondents with high education and digital literacy, when faced with breastfeeding problems, tended to seek advice and solutions from social media (Internet and Facebook pages) rather than through health personnel and lactation clinics. This implies that current counselling services for lactating mothers are not convenient or responsive [42,43]. Hence, further study is needed to evaluate and review existing health services and guidelines related to breastfeeding support, as well as the compliance with the BFHI recommendations.

However, the pattern of mothers using the Internet to search for breastfeeding information in this study is interesting and similar to mothers in the Northeast of Thailand [23]. Taking into consideration maternal lifestyles in urban settings and the increasing trend of Internet and social media use in Thailand [44], online counselling and training on breastfeeding can be an important, emerging platform to provide problem-solving and peer support for all lactating mothers, given the acceptable validity of the information provided. Thus, policy makers should capitalize on the advantages, and potential, of online technology to reach out to more lactating mothers.

This study contains both strengths and limitations. For strengths, first, this study is one of the first papers to explore current breastfeeding prevalence, common practices and problems, and the potential contributing factors for 6-month EBF in Bangkok. Second, the multivariate analysis using mixed-effects models is useful in correcting the influence of a hierarchical structure derived from different types of health facilities.

However, some limitations remained. First, some degree of selection bias could have occurred since the participants were recruited when they accessed the well-child clinic in selected health facilities, and mothers who did not accompany their children to the selected sites were excluded by default. In addition, we were not able to recruit mothers from private hospitals despite our attempts in contacting them. Thus, mothers who were able to afford the cost of services at private facilities (probably richer mothers) were not covered by this study. Second, recall bias could have occurred as some mothers may not have been able to remember the day that they first introduced water, breastmilk substitutes, complementary foods, or processed supplementary foods to their babies. Third, it is likely that some key variables were not collected and analyzed in this study, especially socio-cultural determinants such as exposure to advertisements on breastmilk substitutes or social support in communities. An interview regarding breastfeeding practice undertaken at participants’ households is recommended to prevent the selection bias from acquiring only mothers who are able to access facility-based healthcare, though this approach is costly.

## 5. Conclusions

The study clearly demonstrates that mothers living in Bangkok still practice EBF at a suboptimal rate (41%) and, given the WHO recommendation for 6-month breastfeeding of 50% by 2025, more work needs to be done for Thailand to reach its EBF target. In the univariable analysis, the key determinants that influenced 6-month EBF included: a maternal age of more than 30 years; higher education level; higher maternal income (THB 13,000 or more per a month); being multiparous; exposure to breastfeeding advice during pregnancy; intention to breastfeed for a long duration during pregnancy; experience of 6-month EBF in the previous child. In multivariate analysis, the intention to breastfeed for a long duration, stated during pregnancy, was associated with 6-month EBF (Odds Ratio [OR] 2.9, 95% Confidence Interval [CI] 1.7–4.9). With a focus on mothers with two or more children only, the intention to perform breastfeeding for a long duration stated during pregnancy (OR 3.1, CI 1.3–7.3) and experience of 6-month EBF in the previous child (OR 2.1, CI 1.1–4.1) were strongly associated with 6-month EBF in the current child.

This study suggests that policy makers should strengthen maternal health services to support and empower all expectant mothers during pregnancy and the lactating period to develop and maintain their strong intention to breastfeed for at least six months through intensive counselling and training programs, with additional care for mothers with a high risk of failure to achieve EBF. Additionally, the application of online technology to breastfeeding related services could increase access to breastfeeding counselling and support for lactating mothers. Further studies should focus on an interventional study to promote maternal intentions to breastfeed, complemented with a qualitative approach to investigate practical experiences in achieving EBF.

## Figures and Tables

**Figure 1 ijerph-18-07889-f001:**
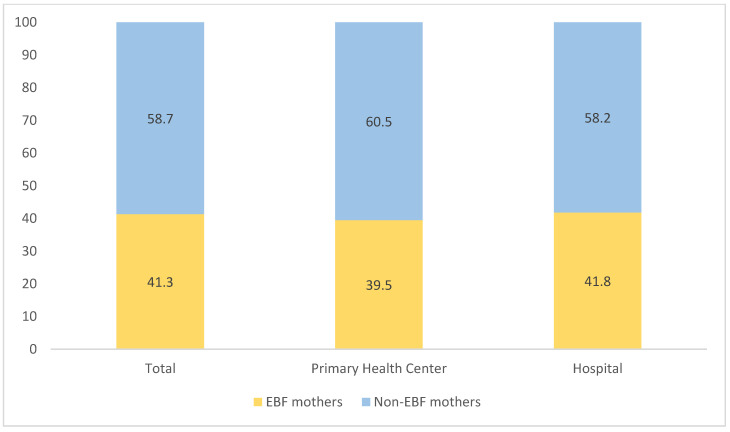
The percentage of participants by types of health facility and EBF outcome.

**Figure 2 ijerph-18-07889-f002:**
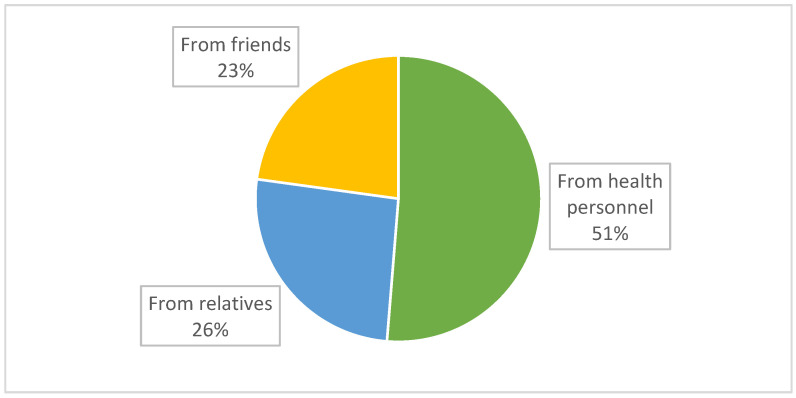
Sources of breastfeeding advice during pregnancy.

**Figure 3 ijerph-18-07889-f003:**
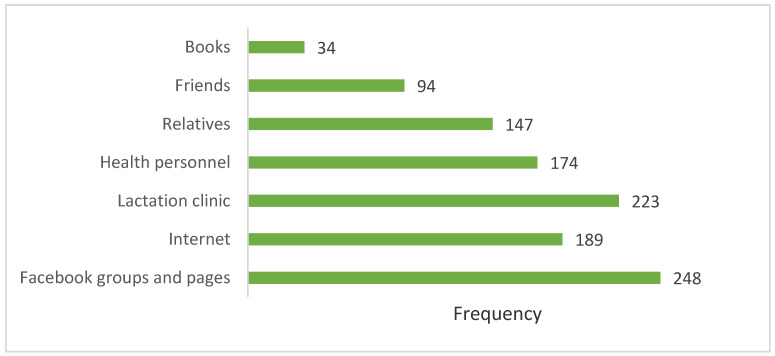
Sources of support/advice when mothers faced breastfeeding difficulties.

**Figure 4 ijerph-18-07889-f004:**
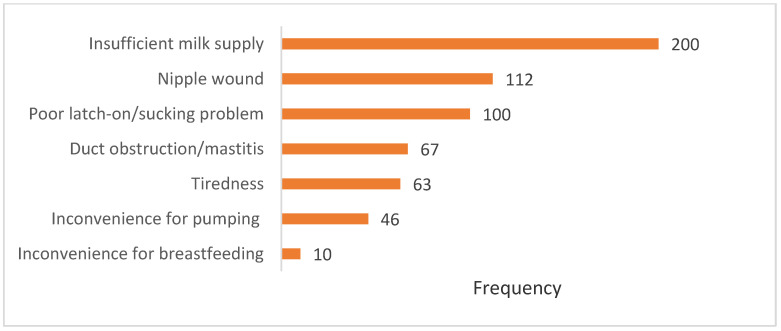
Breastfeeding problems.

**Figure 5 ijerph-18-07889-f005:**
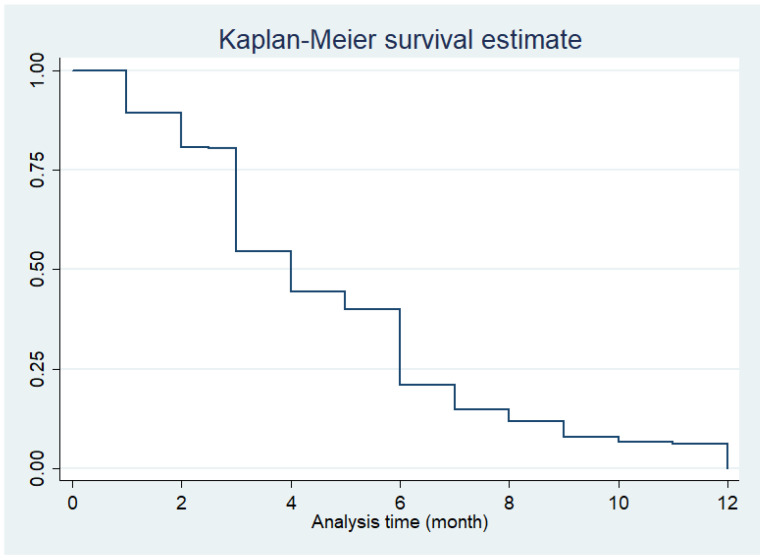
Six-month exclusive breastfeeding survival analysis (months).

**Table 1 ijerph-18-07889-t001:** Comparing the six-month exclusive breastfeeding (EBF) by personal attributes.

Variables	Overall Mothers (%)	EBF Mothers (%)	Non- EBF Mothers (%)
Age			
Median (p25, p75)	30 (24, 35)	31 (26, 35) **	29 (23, 34) **
Age group			
<30 years	331 (49.7)	119 (43.3) **	212 (54.2) **
≥30 years	335 (50.3)	156 (56.7) **	179 (45.8) **
Total	666 (100.0)	275 (100.0)	391 (100.0)
Education level			
Secondary school or below	400 (59.5)	151 (54.7) *	249 (62.9) *
Diploma/Bachelor degree or above	272 (40.5)	125 (45.3) *	147 (37.1) *
Total	672 (100.0)	276 (100.0)	396 (100.0)
Working status			
Not working	254 (39.3)	113 (42.6)	141 (36.9)
Working	393 (60.7)	152 (57.4)	241 (63.1)
Total	647 (100.0)	265 (100.0)	382 (100.0)
Maternal income (Baht/month)			
Median (p25, p75)	13,000 (9000, 20,000)	15,000 (9000, 25,000)	12,000 (9000, 20,000)
Maternal income range			
<13,000 Baht/month	250 (48.9)	87 (43.1) *	163 (52.8) *
≥13,000 Baht/month	261 (51.1)	115 (56.9) *	146 (47.2) *
Total	511 (100.0)	202 (100.0)	309 (100.0)
Marital status			
Always living with husband	574 (85.3)	242 (87.0)	332 (84.0)
Living with husband for sometimes/living alone	99 (14.7)	36 (13.0)	63 (16.0)
Total	673 (100.0)	278 (100.0)	395 (100.0)
Number of children			
1	343 (50.9)	120 (43.3) ***	223 (56.2) ***
≥2	331 (49.1)	157 (56.7) ***	174 (44.8) ***
Total	674 (100.0)	277 (100.0)	397 (100.0)
Experience of 6-month EBF in the previous child ^+^			
No	84 (26.8)	25 (16.4) ***	59 (36.9) ***
Yes	229 (73.2)	128 (83.7) ***	101 (63.1) ***
Total	313 (100.0)	146 (100.0)	160 (100.0)
Exposure to breastfeeding advice during pregnancy			
No	121 (18.3)	38 (13.9) *	83 (21.3) *
Yes	541 (81.7)	235 (86.1) *	306 (78.7) *
Total	662 (100.0)	273 (100.0)	389 (100.0)
Intention to breastfeed during pregnancy			
For a short duration (<6 months)	146 (22.3)	29 (10.7) ***	117 (30.4) ***
For a long duration (≥6 months)	510 (77.7)	242 (89.3) ***	268 (69.6) ***
Total	656 (100.0)	271 (100.0)	385 (100.0)
Mode of delivery			
Normal labor	397 (59.4)	162 (58.7)	235 (60.0)
Caesarean section or labor assisted	271 (40.6)	114 (41.3)	157 (40.0)
Total	668 (100.0)	276 (100.0)	392 (100.0)
Rooming-in during the hospital stay			
No/Room-in for some time	278 (41.6)	123 (44.6)	155 (39.5)
Yes	390 (58.4)	153 (55.4)	237 (60.5)
Total	668 (100)	276 (100.0)	392 (100.0)
Exposure to breastfeeding problems			
No	283 (43.5)	123 (46.6)	160 (41.4)
Yes	367 (56.5)	141 (53.4)	226 (58.6)
Total	650 (100.0)	264 (100.0)	386 (100.0)

* A *p* value < 0.05, ** A *p* value < 0.01, *** A *p* value < 0.001. ^+^ for multiparous mothers only.

**Table 2 ijerph-18-07889-t002:** Period of exclusive breastfeeding (EBF) survival by personal attributes (log rank).

Variables	Median (p25, p75)	*p* Value
Age group		
<30 years	4 (3, 6)	0.014 *
≥30 years	4 (3, 7)	
Education level		
Secondary school or below	3 (3, 6)	0.009 **
Diploma/Bachelor degree or above	5 (3, 7)	
Working status		
Not working	3 (3, 6)	0.221
Working	4 (3, 6)	
Maternal income range		
<13,000 Baht/month	4 (3, 6)	0.269
≥13,000 Baht/month	5 (3, 6)	
Marital status		
Living with husband	4 (3, 6)	0.214
Living with husband for sometimes/living alone	3.5 (3, 6)	
Number of children		
1	4 (3, 6)	0.205
≥2	4 (3, 6)	
Experience of 6-month EBF in the previous child		
No	3 (2, 5)	<0.001 ***
Yes	6 (3, 7)	
Exposure to breastfeeding advice during pregnancy		
No	3 (2, 6)	0.508
Yes	4 (3, 6)	
Intention to breastfeed during pregnancy		
For a short duration (<6 months)	3 (3, 4)	<0.001 ***
For a long duration (≥6 months)	6 (3, 7)	
Mode of delivery		
Normal labor	4 (3, 6)	0.856
Caesarean section or labor assisted	4 (3, 6)	
Rooming-in during the hospital stay		
No/Room-in for sometimes	4 (3, 6)	0.888
Yes	4 (3, 6)	
Exposure to breastfeeding problems		
No	4 (3, 6)	0.264
Yes	4 (3, 6)	

* A *p* value < 0.05, ** A *p* value < 0.01, *** A *p* value < 0.001.

**Table 3 ijerph-18-07889-t003:** Six-month exclusive breastfeeding (EBF) by personal attributes: multivariate analysis (full N).

Variables	Multivariate Analysis (Mixed-Effects Multi-Level Model)	
	Adjusted Odds Ratio	95% Confidence Interval
Age groups		
≥30 years	1.5	1.0–2.2
(ref = <30 years)		
Education level		
Diploma/Bachelor degree or above	1.3	0.9–2.1
(ref = secondary school or below)		
Maternal income range		
≥13,000 Baht/month	1.3	0.8–2.0
(ref = <13,000 Baht/month)		
Number of children		
≥2	1.3	0.8–1.9
(ref = 1)		
Exposure to breastfeeding advices during pregnancy		
Yes	1.3	0.8–2.1
(ref = no)		
Intention to breastfeed during pregnancy		
For a long duration (≥6 months)	2.9 *	1.7–4.9
(ref = or a short duration (<6 months))		

* A *p* value < 0.001.

**Table 4 ijerph-18-07889-t004:** Six-month exclusive breastfeeding (EBF) by personal attributes: multivariate analysis (partial N).

Variables	Multivariate Analysis (Mixed-Effects Multi-Level Model)	
	Adjusted Odds Ratio	95% Confidence Interval
Age groups		
≥30 years	1.3	0.7–2.4
(ref = <30 years)		
Education level		
Diploma/Bachelor degree or above	1.4	0.7–2.7
(ref = secondary school or below)		
Maternal income range		
≥13,000 Baht/month	1.4	0.8–2.6
(ref = <13,000 Baht/month)		
Experience of 6-month EBF in the previous child		
Yes	2.1 *	1.1–4.1
(ref = no)		
Exposure to breastfeeding advices during pregnancy		
Yes	1.0	0.5–2.2
(ref = no)		
Intention to breastfeed during pregnancy		
For a long duration (≥6 months)	3.1 *	1.3–7.3
(ref = or a short duration (<6 months))		

* A *p* value < 0.05.

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
