# Peer review of "Breastfeeding Practice and Association between Characteristics and Experiences of Mothers Living in Bangkok"

_ijerph, 2021, doi:10.3390/ijerph18157889_

Round 1

Reviewer 1 Report

It has been a pleasure to review this manuscript.
The benefits of exclusive breastfeeding are known throughout the world, but exclusive breastfeeding rates are below desirable levels.
This research explores the prevalence of breastfeeding in the first 6 months from birth in Bangkok.
I think the introduction provides enough information.
The design and methodology of the study also seem correct to me.
The strong point of this research is in the associations between the different variables of the study and the mixed effects model used in the multivariate analysis.
The discussion seems correct to me, the results are correctly discussed and the limitations that this research could have are recognized.
The conclusions are supported by the results.
Honestly, I think a good job was done.

Thanks

Regards

Author Response

Dear reviewer,

Thank you so much for your kind comments. We are so grateful for your support. Please see the revised manuscript that has been improved in accordance with the suggestions by other two reviewers in the clean text.

Reviewer 2 Report

A very good presentation of an equaly interesting research. Good, easy to follow stucture, detailed description of methodology and statistical analysis. 

It is a well written manuscript but some changes must be done. The authors should taken into account the journal's guidelines about the abstract's form and the tables' style. 

Also, the text of discussion is big. In my opinion, authors must improve this text. They should re-write some paragraphs, remove some sentences, etc. This part of manuscript is very important and it must be more clear.    

Author Response

Dear reviewer,

Thank you so much for your thorough reading and many useful comments for revision. We have edited the manuscript according to all of your suggestions, please see our point-by-point responses to your comments as following;

  1. We have edited the abstract to be within 200 word counts and followed the style of structured required by the journal’s guidelines. Also, all tables were checked on their formats and styles.
  2. The discussion part has been revised in every paragraph to be more precise, we have deleted all redundant section. please kindly see the discussion section in line 280-357.

We submitted two document for your consideration.

  1. Revised manuscript in track change form.
  2. Revised manuscript in clean text.

Please note that the line number that we mentioned in our responses is referred to the line number in the revised manuscript in clean text. Please let us know if you have more comments and suggestions.

Best wishes,

Chompoonut

Reviewer 3 Report

Abstract:

L#19 mentions national statistics instead of mentioning few mothers in Thailand exclusively breastfeed their children.....

Line#21 I would suggest writing as below

prevalence of exclusive breastfeeding up to six months..... Moreover, I would suggest just say exclusive breastfeeding instead of mentioning up to six months again and again. Just mention this in the method section that researchers define exclusive breastfeeding--the child who exclusively breastfeed up to six months.

Introduction

I would suggest revising the introduction section in a more organized way. For example, the third para of the intro discusses the factors associated with EBF, and similar findings are mentioned in the abstract section. The question arises here that if all the factors are known before this research, then how this research adds value to existing literature. Explain study gaps in the introduction section and then present a novel approach to address that knowledge gap through this research.

Method section

L#125-139 can be labeled as Questionnaire Development

The data collection heading can be started from L#141

Moreover, the word "Data" is plural, so use were instead of mentioning was 

What are C.T and T.T?? Explain it

 L#150=156. The authors explained that they measured EBF with three questions with options such as: have including water, breastmilk substitutes, complementary foods, and processed supplementary foods.

How authors indexed these three questions and explain the outcome variable was a dichotomous variable or having more than two categories. And explain the whole procedure of indexing the variable.

L#167-177 can be mentioned a separate headline such as "data analysis"

L#167-- There is no satistical test such as "Chi-square test regression". I think the authors wanted to mention regression analysis. Please check and make corrections. 

L#168 multivariable??? write complete name of test...

L3175-176. It would be good if authors mention software in starting lines of analysis.

Ethical consideration

Mention the consent procedure and stipend which researchers give to study participant in this section.

Results:

Figure 1

Just mention 'exclusive breastfeeding, and not exclusive breastfeeding

and explain data in percentage instead of frequencies

L#(p25-p75 = 24-35)... What is P24-p75?

Table1

Don't mention the Total row for each variable. Just mention your sample size after the end of the heading of table 1

Figure 04

Why authors presented findings in frequency instead of percentage?

Discussion

The discussion section is fine. However, the authors can mention the strength and limitations of the study at the end of the discussion section.

Conclusions

The findings of the conclusions and abstract are the same. the authors can discuss policy implications in the conclusion section 

Round 2

Reviewer 3 Report

I have carefully reviewed the manuscript and authors' replies to each point. I  recommend accepting this manuscript in its present form.